

# A first estimation of the contraction related to vertical axis rotation: the case of the Ibero-Armorican Arc formation

Josep Maria Casas[1]*, Joan Guimerà[1-2], Joaquina Alvarez-Marron[3], Ícaro Días da Silva[4-5]

[1] *Dpt. de Dinàmica de la Terra i de l'Oceà, Universitat de Barcelona, Martí Franquès s/n, 08028 Barcelona, Spain, casas@ub.edu, joan.guimera@ub.edu*

[2] *Institut de recerca GEOMODELS, Martí Franquès s/n, 08028 Barcelona, Spain.*

[3] *Dpt. of Earth Structure and Dynamics, and Crystallography, Institute of Earth Sciences, Jaume Almera, CSIC, Lluís Solé i Sabarís s/n, 08028 Barcelona, Spain, jalvarez@ictja.csic.es*

[4] *Instituto Dom Luiz (IDL), Faculdade de Ciências, Universidade de Lisboa, Campo Grande, Edif. C1, Piso 1, 1749-016 Lisboa, Portugal, ipicaparopo@gmail.com*

[5] *Dpto. de Geologia, Faculdade de Ciências, Universidade de Lisboa, Campo Grande, 1749-016 Lisboa, Portugal, ipicaparopo@gmail.com*

\* *Corresponding author (e-mail: casas@ub.edu)*

**Abstract.** Different models have been proposed to explain the formation of the Ibero-Armorican Arc, which require significant vertical axis rotations, at the end of the Variscan orogeny. Estimates of the amount of contraction (horizontal shortening) needed for these rotations range from 54% to 91% perpendicularly to the arc. These estimates are compared with coeval deformational structures developed in two areas of the orogen, one in the autochthonous hinterland underlying the Galicia-Trás-os-Montes Zone in the southern branch of the arc, and the other in the Cantabrian Zone foreland in the core of the arc. From this analysis it follows that the late Variscan deformation together with the subsequent Alpine contraction is not sufficient to explain the formation of the Ibero-Armorican Arc as a secondary structure by means of vertical axis rotations. Our analysis suggests this arc is mainly a primary, or non-rotational curve, slightly modified by ca. 10% of superposed contraction during late Carboniferous and/or Alpine times. Moreover, we propose that the assumptions underlying the interpreted geometry of the arc be re-evaluated, and we discuss the role of late-Variscan regional strike-slip faults in the Iberian and in the Armorican massifs that probably acted consecutively before and during the contraction of the arc.

## 1. Introduction

Different models have been proposed to synthesize the structural evolution and explain the characteristic arcuate geometry of the western European Variscan Belt, known as the Ibero-Armorican Arc (IAA, Fig. 1). The indentor model requires a curved geometry of the Gondwanan margin prior to collision and highlights the role of the Gondwana promontory during the Gondwana-Laurussia collision (Matte and Ribeiro, 1975; Lefort, 1979; Brun and Burg, 1982; Burg et al., 1987; Quesada, 1991; Dias and Ribeiro, 1995; Sánchez-García et al., 2003; Simancas et al., 2009) (Fig. 2A). Other models propose that this arc constitutes an orocline formed by secondary vertical axis buckling of an originally oriented N-S belt during Gondwana-Laurussia collision (Weil et al., 2000, 2001, 2010, 2013a, 2013b, 2019; Gutiérrez-Alonso et al., 2012; Fernández-Lozano et al., 2016; Pastor-Galán et al., 2011, 2012b, 2015a, 2017, 2019; Shaw et al., 2012) (Fig. 2B). A third group of models emphasizes the role of the large-scale strike-slip shear zones and the associated deformation as the main origin for the formation of the Ibero-Armorican arc (Martínez-Catalán et al., 2007; Martínez-Catalán, 2011) (Fig. 2C). Although the oroclinal and indentor models



may appear to be mutually exclusive, some authors reconcile both models in a proposal involving a combination of
some indentation and subsequent sinistral and dextral motion along shear zones on either side of the promontory
(Murphy et al., 2016) or subsequent buckling (Casas and Murphy, 2018) (Fig. 2D). One of the most critical points to
discern between the different proposals is the amount of vertical axis rotation required. The secondary orocline
models propose counter-clockwise vertical axis rotations ranging from 60º or 70º to 90º for the southern arm of the
Ibero-Armorican Arc and clockwise rotations of 25º for the northern arm (Pastor-Galán et al., 2015a, b, 2016, 2017).
In contrast, the amount of rotation required for some of the models involving indentation and subsequent buckling is
ca. 27.5º for both branches (Casas and Murphy, 2018). Although they clearly differ, these vertical axis rotations
each require a significant amount of contraction and extension that has yet to be quantified but which should be
recognizable in the surrounding regions. With the exception of the indentor model, the other models agree that the
vertical axis rotations occurred in late Carboniferous-Early Permian times (ca. 305-295 Ma), after the main Variscan
deformational events that include the emplacement of the regional allochthons. In this contribution, we estimate the
amount of contraction required for the different vertical axis rotations proposed for the formation of the Ibero-
Armorican Arc, and we compare the obtained data with deformational structures developed simultaneously. For this
comparison, we focus in two areas in the southern branch of the arc: (i) the parautochthon of the Galicia-Trás-os-
Montes Zone and the underlying autochthonous hinterland of the Central Iberian Zone in the southern branch of the
arc, and (ii) the Cantabrian Zone foreland in the core of the arc. Incorporation of these results into future
reconstructions may test the validity of the proposed models, and constrain the geometry and the origin of the arc
and the role of the large strike-slip faults.
In this contribution we follow the Marshak's (2004) terminology for curving fold-thrust belts which distinguishes
between rotational and non-rotational curvatures depending on whether or not segments of the belt rotated around an
imaginary vertical axis.

## 61    2. Geological setting of the Iberian Massif

The Iberian Massif includes several tectonically juxtaposed geological domains with important differences in their
stratigraphic, structural, magmatic and metamorphic evolution (Lotze, 1945; Julivert et al., 1972; Farias et al., 1987)
(Fig. 1). In SW Iberia, the South-Portuguese Zone (SPZ) is a tectonically imbricated Devonian to late Carboniferous
stratigraphic sequence, facing towards the southwest, that consists of synorogenic marine sediments with abundant
volcano-sedimentary components (Oliveira et al., 2019) with Laurusian affinity (Braid et al., 2011; Pérez Cáceres et
a., 2017; Pereira et al., 2020). The Ossa Morena Zone (OMZ) represents the northern Gondwana margin (Quesada,
1991; Pereira et al., 2012), composed of Neoproterozoic, Cadomian-related, synorogenic sediments unconformably
overlain by lower Cambrian to Lower Devonian passive-margin volcano-sedimentary sequences (Sánchez García et
al., 2019; Gutierrez Marco et al., 2019), which are in turn unconformably overlain by Upper Devonian to early
Carboniferous flysch basins (e.g. Camargo Rocha et al., 2009; Oliveira et al., 2019).
The boundary of the OMZ with the northern sector of the Iberian Massif is the Coimbra-Cordoba Shear Zone (e.g.
Pereira et al., 2008) or the Badajoz-Cordoba Shear Zone (BCSZ) (Azor et al., 1994, 2019) (Fig. 1). The northern
Iberian Massif can be divided in two major domains: i) The Allochthon, called Galicia-Tras-os-Montes Zone
(GTMZ) includes several stacked sheets; ii) the Autochthon, comprising the Central Iberian (CIZ, hinterland), West
Asturo-Leonese (WALZ) and the Cantabrian Zone (CZ, foreland) in the core of the Iberian-Armorican Arc (e.g.
Lotze, 1945; Julivert et al., 1972; Farias et al., 1987; Gutierrez Marco et al., 1990; Pérez-Estaún et al., 1988, 1990).
The GTMZ show increasing tectonic transport from the lowest to the highest allochthon. In the highest tectonic



slices, the Upper Allochthon represents a terrane that was detached from the northern Gondwanan margin during the
Lower Palaeozoic opening of the Rheic Ocean, accreted to southern Laurussia in the Silurian and in the Devonian
during Variscan collision was thrust over the Iberian margin of Gondwana (e.g. Gómez Barreiro et al., 2007;
Martínez Catalán et al., 2019). The Middle Allochthon is a rootless suture zone that preserves vestiges of the Rheic
Ocean that was consumed by subduction beneath the southern Laurussian margin (e.g. Arenas and Sánchez
Martínez, 2015). The Lower Allochthon represents a segment of the Gondwanan margin that was subducted and
incorporated by obduction into the Variscan accretionary prism (e.g. Díez Fernández et al., 2011). The lowermost
tectonic unit of the GTMZ, the Parautochthon (Dias da Silva et al., 2014a, 2015, 2020; González Clavijo et al.,
2016) also called the Schistose Domain (Farias et al., 1987) or the Parautochthonous Thrust Complex (Ribeiro et al.,
1990), can be divided into Upper Parautochthon, made up of a Cambrian-Silurian stratigraphic sequence comparable
to the CIZ (Dias da Silva et al., 2014a, 2015, 2016); and the Lower Parautochthon comprised of synorogenic early
Carboniferous marine strata imbricated in a piggy-back thrust sequence (Rodrigues et al., 2013; Dias da Silva et al.,
2015; González Clavijo et al., 2016; Martínez Catalán et al., 2016).
The autochthon consists of Neoproterozoic to Lower Devonian stratigraphic sequences, within which are two major
unconformities that record global Lower Palaeozoic extensional events related to the formation of a passive margin
along the northern margin of Gondwana and the opening of the Rheic Ocean (Gutierrez Marco et al., 1990; Martínez
Catalán et al., 1992; Dias da Silva et al., 2011; 2014b). The WALZ and CZ preserve more proximal facies than the
CIZ (Marcos et al., 2004).
The deformation history of the Iberian Massif is complex, polyphase and diachronous. Devonian deformation (ca.
410-370 Ma) reflects subduction-related metamorphism followed by obduction, development of detachments and
out-of-sequence thrusting recognized in the allochthonous complexes of the GTMZ (e.g. Gómez Barreiro et al.,
2007; Martínez Catalán et al., 2009). This deformation is younger in the Lower Allochthon and older in the Upper
Allochthon, and records the orogenic stacking and progradation of the orogenic front towards the Gondwana
foreland.
In northern Iberia, the oldest Variscan deformation and metamorphic event started in the Upper Parautochthon and
CIZ (365-355 Ma) and migrated into the WALZ and CZ by about 340-310 Ma (Dallmeyer et al. 1997). This $D_1$-$M_1$
stage was accompanied by the development of an $S_1$ axial planar cleavage and by chlorite to biotite zone
metamorphism that was synchronous with the development of a foreland basin fed by detritus derived from both the
accretionary prism and the peripheral bulge in the autochthon (Dias da Silva et al., 2015). Around 340 Ma
(Dallmeyer et al., 1997), the Upper Parautochthon was tectonically imbricated at the base of the unrooted
allochthonous complexes (Martínez Catalán et al., 2009) and both were tectonically transported towards present-day
southeast (Dias da Silva, 2014; Dias da Silva et al., 2020). The Upper Parautochthon was thrust onto its foreland
basin (Lower Parautochthon) (Dias da Silva et al., 2014a, 2015). This stage ($C_2$ after Martínez Catalán et al., 2014)
was responsible for the underthrusting of the underlying autochthon, leading to the regional Barrovian
metamorphism peak ($M_1$). The emplacement of the GTMZ deformed the precursor $D_1$ folds ($C_1$ after Martínez
Catalán et al., 2014), and caused the folding of rocks in both Parautochthon and the underlying autochthon around
the basal thrust-zones of the GTMZ (Ribeiro, 1974; Dias da Silva, 2014; Pastor-Galán et al., 2019; Dias da Silva et
al., 2020) thus forming the Central Iberian Arc according to Martínez Catalán et al. (2014) and Dias da Silva et al.
117  (2020).

Tectonic stacking of the GTMZ triggered synorogenic extension and adiabatic decompression ($D_2$-$M_2$) ($E_1$ in
Martínez Catalán et al., 2014 and Alcock et al., 2015). The $D_2$-$M_2$ event (340-320 Ma) is characterized by the
formation of extensional gneiss domes with orogen-parallel transport of the hanging-wall lithologies towards the



southeast (modern coordinates) (Escuder et al., 1994; Arenas and Martínez Catalán, 2013; Díez Fernández and
Pereira, 2016; Rubio Pascual et al., 2016). The gneiss domes developed in northern Iberia beneath the GTMZ, are
rooted in the autochthon (Díez Fernández et al., 2017). The $D_2$-$M_2$ event progressed towards more continental
realms, affecting the WALZ at about 320-310 Ma (Martínez Catalán et al., 2003). Synchronously, the orogenic front
migrated further into Gondwana, with the development of the foreland thrust belt in the CZ (Pérez-Estaún et al.,
126 1988).

The late Variscan (ca. 315-300 Ma) in Iberia is marked by heterogeneous upright folding and transcurrent brittle-
ductile shear zones, under low-grade metamorphic conditions ($D_3$-$M_3$) (Gonzalez Clavijo et al., 1993; Gutierrez-
Alonso et al., 2015; Díez Fernández and Pereira, 2016, 2017; Dias da Silva et al., 2018). The $D_3$-$M_3$ folds folds,
cogenetic stretching lineations and axial planar cleavages, are parallel to the orogenic trend and with the $D_1$-$M_1$ folds
(Pastor-Galán et al., 2019). At the end of the $D_3$, deformation became focused into a network of brittle-ductile shear
zones, affecting especially the margins of $D_2$-$M_2$ gneiss dome cores and pre- to late-$D_3$ granitic intrusions (Fig. 1).
Conjugate dextral and sinistral shear zones locally steepened the previous fabrics, causing localized retrograde
metamorphism, and shuffled the different tectono-metamorphic domains, juxtaposing low grade and high grade
metamorphic rocks (e.g. Díez Fernández and Pereira, 2016; Dias da Silva et al., 2020). During this stage (307-300
Ma) the Porto-Tomar Shear Zone (PTSZ, Fig. 1) formed (Gutiérrez-Alonso et al., 2015) as a major dextral strike-
slip shear zone that, according to some authors, connects with the dextral Armorican shear zones in the Armorican
Massif (Martínez-Catalán et al., 2007; Martínez-Catalán, 2011, 2012) (Fig. 1).
**3. Proposed models for the arc formation**
The models that propose a secondary origin for the Ibero-Armorican Arc (IAA) agree that it formed at the end of the
$D_3$ deformational event (305-295 Ma, Kasimovian-Asselian) (Weil, 2006; Weil et al., 2010; Martínez-Catalán, 2012)
and in a geometry that resembles a vertically-plunging fold. However, they assign different mechanisms to its
formation (Fig. 2B and C). A group of models argue that the arc formed as a result of a lithospheric-scale buckling
of an initial N-S oriented linear orogen (Weil et al., 2000, 2001, 2010, 2013a, 2013b, 2019; Gutiérrez-Alonso et al.,
2004, 2012; Fernández-Lozano et al., 2016; Pastor-Galán et al., 2011, 2012b, 2015a, 2017, 2019; Shaw et al., 2012).
This buckling caused a rotation of ca. 90º of both arms of the arc, giving rise to a vertical fold with a tight geometry
(Fig. 2B). This model implies a 90º change in the contraction direction (from E-W to N-S) during the Variscan
deformation. The model has other geodynamic implications, such as the development of an important magmatic
event related to delamination of the thickened lithospheric root at the core of the arc (Gutiérrez-Alonso et al., 2011).
Structures accounting for the deformation related to these important vertical axis rotations have been described only
in the Cantabrian Zone, in the core of the arc, where the development of a system of radial folds simultaneously with
the arc formation, together with the reactivation of existing thrust sheets (Esla Unit) and the southward thrusting of
the Picos de Europa Unit have been proposed (Weil et al., 2000, 2001, 2013a, 2013b; Pastor-Galán et al., 2012a;
Merino-Tomé et al., 2009).
Martínez-Catalán et al. (2007) and Martínez-Catalán (2011) have suggested that the arc formed as a result of
symmetrical 90º rotation of existing tectonostratigraphic zones due to continental-scale dextral shear faulting
originated by an oblique collision. As with the previous model, this rotation resulted in an arc with tight geometry
(Fig. 2C). In this model, the Porto Tomar shear zone (Ribeiro et al., 1980) merges into the South Armorican shear
zone (Shelley and Bossière, 2002; Martínez-Catalán et al., 2007) in the Armorican Massif. These faults were
initially quasi-linear and significant changes in their orientation (NE-SW in the western margin of the Iberian



Massif, E–W in northwest Galicia and WNW-ESE in the Armorican Massif) were caused by folding during the
formation of the Ibero-Armorican Arc. According to Martínez-Catalán (2011) the motion of these dextral shear
zones during Gondwana–Laurussia convergence explains the stratigraphic similarity between the Central Armorican
and the Central Iberian zones and may account for the original proximity of Crozon (western Armorican Massif) and
Buçaco (western Portugal) (Fig. 1) which exhibit similar Ordovician stratigraphic successions (Young, 1988, 1990;
Robardet, 2002 and references therein). This proposal implies a southern original position for the Armorican Massif
either to the south or in front or of the western edge of the Iberian Massif during the Ordovician, before the
development of the IAA (Young, 1990; Robardet, 2002). A southernmost position for the Armorican Massif during
the Ordovician has also been proposed to explain the distribution of the Late Ordovician (Katian) Nicolella
Community brachiopod populations by Colmenar (2015).
Other group of models invoke a combination of indentation and subsequent dextral and sinistral motion along shear
zones on either side of the promontory (Murphy et al., 2016) or subsequent buckling (Casas and Murphy, 2018).
Murphy et al. (2016) propose that the continental edge can have a promontory, even if the geological belts are
approximately-linear and the model of Casas and Murphy (2018) is based on palinspastic restoration and pre-
orogenic geological constraints and proposes a Gondwanan margin with an irregular pre-Variscan geometry, with
two E-W oriented segments linking a N-S central segment. This Gondwanan promontory was cut and offset by
regional strike-slip faults, and each segment was rotated ca. 27.5° about a vertical axis during late Variscan arc
formation (Fig. 2D). This amount of rotation has been deduced from the geometry of the South Armorican and
Porto-Tomar shear zones, assuming that they initially constituted the same fault system with a linear geometry (Fig.
2D).
Some authors propose the existence of a second arc in the Iberian Massif, with opposite curvature to the IAA, the
Central Iberian Arc (CIA) (Fig. 1). However, there is great controversy about its age of formation or even its actual
existence. According to Martínez-Catalán et al. (2014) and Dias da Silva et al. (2020), the CIA was a consequence
of the tectonic imbrication of the GTMZ onto the Iberian autochthon. This imbrication would have caused the
rotation of the previous structures and stratigraphy, that was tightened during the $D_3$ stage, synchronously with the
formation of the IAA. In contrast, Shaw et al. (2012, 2014), Gutierrez-Alonso *et al.* (2015) and Weil et al. (2019)
have proposed that the formation of IAA and CIA arcs was contemporaneous, due to oroclinal bending of a former
linear orogen with a N-S trend, taking place at the end of the Variscan orogeny. However, Pastor-Galán et al.
(2015a, 2016, 2017, 2019) have argued that the CIA is an inherited feature that supports the pre-Variscan irregular
geometry of the Gondwanan margin proposed by Casas and Murphy (2018). Pastor-Galán et al. (2015a) proposed
that it had to have formed prior to the development of the IAA, even if the CIA were a secondary arc. Dias da Silva
et al. (2020) added that the secondary origin of the CIA is related to the thin skinned emplacement of a lateral
extrusion wedge formed by the collapse of the Variscan accretionary prism in the French Massif Central at
approximately 360-340 Ma.
**4. The amount of contraction required by the different proposed models**
A simplified way to estimate the amount of contraction required to cause rotation along the vertical axis of the
branches of the arc is presented in Figure 3. We estimate the amount of contraction related to the different vertical
axis rotations, which has been proposed for the southern branch of the IAA, by comparing the initial length of a
segment of the arc versus the final width that separates the tip of the two branches of the arc defined by the same
segment (Fig. 3). In doing so, we obtain the percentage of contraction between the two arms of the arc. The

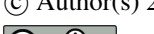



contraction estimate obtained depends on the original length of the previously linear orogen and on the final shape
(more or less tight) of the arc. Moreover, we can estimate the amount of "lost" lithosphere by comparing the initial
trace of the WALZ-CZ boundary versus its final trace (Figs. 4 and 5).
The oroclinal bending about a vertical axis of a linear orogen (Weil et al., 2000, 2001, 2010, 2013a, 2013b, 2019;
Gutiérrez-Alonso et al., 2012; Fernández-Lozano et al., 2016; Pastor-Galán et al., 2011, 2012b, 2015a, 2017, 2019;
Shaw et al., 2012) implies a rotation of ca. 90º for both branches of the arc to obtain its tight geometry (Fig. 4A).
After this rotation, a fragment of the linear orogen of at least 1475 km length, exhibits a final length that is
equivalent to the width of the vertical fold, that is 127 km. This implies a contraction of about 91% at the tip of the
inner part of the generated arc (Fig. 4A), and an amount of lost lithosphere around 698 $10^3$ km² (Fig, 5A).
A similar result was obtained in the model of the secondary fold forming as a result of strike-slip faulting (Martínez-
Catalán et al., 2007; Martínez-Catalán, 2011). In this case, considering an initial length of ca. 1550 km and a final
width measured across the fold about 95 km, the amount of required contraction is about 94% and the surface of lost
lithosphere is around 767 $10^3$ km² (Fig. 5B). Moreover, this model implies an important phase of extension in the
hinterland situated north of the arc. We estimate the amount of extension required to be ca. 100% (680 to 1370 km,
Fig. 4B), however features associated with it have not been identified.
The model proposed by Casas and Murphy (2018) requires a rotation of 27.5º for both branches, similar to the ca.
25º clockwise rotation proposed for the northern arm of the arc by Pastor-Galán et al. (2015b). If we apply the same
method to the geometry proposed by Casas and Murphy (2018), the amount of contraction is considerable, ca. 54%,
and the reduction in the surface of lithosphere, although less than in the previous models, is also significant, around
36 $10^3$ km² (Fig. 5C). All the previous figures on the amount of lost lithospheric surface should be considered as
minimum values, as the calculations of Fig 5 assumed no change in the position of the orocline hinge during its
development.
**5. The geological data**
To compare these estimated amounts of contraction with actual structures developed in late Variscan times, we
focussed our analysis in the autochthon of the CIZ, just beneath the parautochthon of the GTMZ, where $D_3$
structures are well developed (Dias da Silva, 2014), and in the Cantabrian Zone, the core of the arc, where a
development of radial folds (Julivert and Marcos, 1973; Weil et al., 2000, 2001, 2012, 2013; Pastor-Galán et al.,
2012a), the reactivation of previous thrusts (Weil et al., 2013) and the emplacement of the youngest thrust units, the
Cuera Unit and the Picos de Europa Province (Merino-Tomé et al., 2009) coeval with the arc formation have been
proposed.
**5.1 The Parautochthon of the GTMZ and the autochthon (CIZ) beneath**
In NW Iberia, $D_3$ structures affect the GTMZ (including the Parautochthon) and the tectonically underlying CIZ
(Fig. 6A). These $D_3$ structures overprint all the previous structures, including the $D_1$ folds that wrap around the
GTMZ, the main basal thrusts and detachments that limit the allochthonous domains as well as the $D_2$ extensional
detachments and $M_2$ isograds that bound the gneiss dome (Azor et al., 2019; Dias da Silva et al. 2020).
There are two types of $D_3$ structures: i) fan-like heterogeneous folds whose style adapted to the previous structures,
creating a strain shadow with convergent folds to the east and beneath the GTMZ, which acted as a rigid body
(Alonso and Rodríguez, 1981; Dias da Silva, 2014); and ii) conjugate dextral and sinistral transcurrent brittle-ductile
shear zones (e.g. González Clavijo et al., 1993; López-Plaza and López-Moro, 2004; Gutierrez Alonso et al., 2015).



The shear zones predominantly affect the pre- to syn-$D_3$ granites and the $D_2$ gneiss dome cores in both the CIZ and
GTMZ (Escuder et al., 1994; Alcock et al., 2015; Díez Fernández et al., 2017; Dias da Silva et al., 2020).
In the eastern rim of GTMZ (i.e. the southern limb of the IAA), the $D_3$ folds range in orientation from oblique to
orthogonal relative to the $D_1$ structures. The interference of $D_3$ folds with $D_1$ and $D_2$ folds formed type 1 and type 2
fold interference patterns of Ramsay and Huber (1987) (see Pastor-Galán et al., 2019; Dias da Silva et al., 2020). As
a consequence, the $D_3$ folds have variable plunges, from vertical to sub horizontal, with WNW and to ESE sense of
plunge and a regular WNW-ESE axial plane. $D_3$ folds exhibit sub-vertical axial surfaces. The axial planar
crenulation cleavage ($S_3$) shows near vertical fan-like dispersion that is more evident where a previous sub-
horizontal layering is present. The folding is highly heterogeneous at all scales, with sectors where there are tight
isoclinal folds separated by areas with more open folds (Dias da Silva et al., 2020).
Retrograde metamorphism and the reworking of the previous fabrics occur adjacent to brittle-ductile transcurrent
conjugate shear zones. These shear zones also produce proto-mylonitic to ultra-mylonitic fabrics associated with
faults, with intrafolial and sheath folds, C- and C'-S pairs, quarter structures, tectonic fish and sigmoidal shapes in
porphyroclasts and syn-tectonic porphyroblasts (as in the aureoles of the syn- to late-$D_3$ granites) (Dias da Silva et
al., 2020). However, in upper structural levels, these features are less common. Instead, a well developed crenulation
cleavage is associated with tight isoclinal folds. Taken together, it appears that $D_3$ flattening was accommodated by
motion along conjugate transcurrent shear zones in the high grade metamorphic domains (gneiss domes, granites,
etc.) and by folding in the low grade metamorphic domains (Dias da Silva et al., 2020).
The $D_3$ folds and shear zones enclose pre-to syn-$D_3$ granitic bodies, which acted as rigid bodies. The $D_3$ structures
steepened some pre-existing structures, such as the thrusts and detachments of the GTMZ and the $D_2$-$M_2$ fabrics and
isograds of the gneiss domes (e.g. Díez-Fernández and Pereira, 2016, 2017; Dias da Silva et al., 2020). The regional
orientation of $D_3$ structures changes from N-S in the western limb of the IAA to WNW-ESE in the southern limb of
the IAA (Martínez Catalán et al., 2014).
The value of contraction caused by the $D_3$ structures in the CIZ was estimated from a cross-section (Fig. 6B). The
base of the Armorican Quartzite was chosen to calculate the contraction in two sectors not affected by granite
intrusions (Fig. 6C). Adding both sectors and comparing their initial and final lengths, the elongation obtained was
equal to -0.05, implying a contraction (horizontal shortening) of 5%. However, the additional contraction produced
by the conjugate, upright, transcurrent brittle-ductile shear zones should be included, although clear markers cannot
be identified to get an estimation, and no significant displacement of the granitic batholiths is depicted in Fig. 6B.
**5.2 The Cantabrian Zone**
The Cantabrian Zone (CZ) is the external thrust-and-fold belt of the Variscan orogen that developed by thin-skinned
deformation at the end of the Carboniferous (Julivert, 1971; Marcos and Pulgar, 1982; Perez-Estaún et al., 1988).
The thrust-and-fold belt deforms pre-orogenic and syn-orogenic Paleozoic sedimentary successions. The youngest
strata of the pre-orogenic succession are Lower Carboniferous in age and form a sedimentary wedge thinning
towards the foreland. The syn-orogenic succession includes late Carboniferous strata and occurs as several clastic
wedges that were deposited in foredeeps coeval with the major thrust events (Marcos and Pulgar, 1982).
The CZ includes a set of arcuate thrust units that were emplaced from west to east in present coordinates. The oldest
thrusts are in the SW and are Bashkirian (Westphalian A) in age (Arboleya, 1981; Marcos and Pulgar, 1982),
whereas the youngest are in the east and are latest Moscovian to Gzhelian in age (Merino-Tomé et al., 2009). The
thrust units include frontal fault-bend and fault-propagation folds developed in response to the ramp-and-flat





geometry of corresponding thrusts (i.e. Bastida et. al., 1984: Alonso, 1987; Pérez-Estaún et al., 1988; Alvarez-
Marron, 1995; Bulnes and Aller, 2002; among others) (Fig. 7, 8). The major folds in the CZ have been classified
into two sets based on the distribution of axial traces in map view with respect to the trend of major thrusts that
impart the arcuate shape to the CZ (Julivert and Marcos, 1973). Folds with axial traces trending sub-parallel to the
trace of the thrusts, mostly developed within individual thrust-sheets, have been classified as longitudinal folds,
whereas folds with axial traces trending sub-perpendicular to the thrusts that extend across several major thrust units
have been classified as radial folds.
More recent studies have shown that lateral structures in the form of ramps and folds and tear faults are common.
These lateral structures may have developed in individual thrust sheets or may reflect the complex evolution and
superposition during emplacement of the thrust pile. The large tectonic superposition of some of the thrust units also
caused accommodation structures in the form of lateral culminations and drop faults on a previously emplaced thrust
sheet above an active underlying sheet. In the CZ, a change in the direction of tectonic transport is particularly
evident between the emplacement of the Picos de Europa and Ponga Units. The former shows transport towards the
NE and E (Alvarez-Marrón, 1995) whereas the latter shows transport towards the S and SSW (Marquínez, 1989).
Most individual thrust-related folds were modified during progressive thrusting and almost every major fold had a
distinctive evolution during the development of the Cantabrian Zone thrust systems. As a result, most folds
originally classified as longitudinal folds are now considered as frontal thrust-related folds (Alonso, 1987, Perez-
Estaún et al., 1988) (Fig. 8) and some folds classified as radial folds are now considered as lateral thrust-related
folds (Alonso, 1987; Bastida and Castro, 1988; Alvarez-Marron, 1995). The Ponga Unit provides a good example of
the variety in distribution, dimensions and attitude of thrust-related folds in the CZ (Fig. 7). The Ponga Unit lies in
the core of the CZ and shows a complex fold interference pattern that was mainly caused by the superposition of
thrust units with different emplacement directions. The emplacement direction was northeastward in rear thrust
sheets but eastward in the frontal thrusts. Most lateral folds with E-W orientation were tightened during the
subsequent south-directed emplacement of the Picos de Europa Unit (Alvarez-Marron, 1995).
The timing of formation of thrust-related folds in the CZ, spans the whole time of development of the Cantabrian
Thrust Systems. The emplacement of earliest thrust units (Somiedo-Correcilla and Esla Units; Arboleya, 1981)
occurred in the late Bashkirian (~ 318 Ma). The south to south-southwest emplacement of the frontal unit in the
internal part of the arc (Picos de Europa Province) occurred during the Kasimovan-Ghezelian transition (~ 304 Ma)
(Merino-Tomé et al., 2009). According to these authors the emplacement of the Picos de Europa Province caused
contraction of 150±15 km that was distributed between southward displacement of the Cuera-Picos de Europa
imbricate thrust system, reactivation of previous thrusts and out-of-sequence thrusting, and by internal deformation.
Part of this contraction also may account for the tightening of the previously formed E-W oriented lateral folds in
the Ponga Unit (Alvarez-Marron, 1995).
Further evidence for the age of formation of longitudinal and radial folds in the CZ is provided by structures
associated with a Late-Variscan extensional episode and metamorphism (Valin et al, 2016). This episode includes
the development of a cleavage that crosscuts longitudinal and radial folds and the frontal thrust of the Picos de
Europa Unit (Aller et al., 1987; Aller et al., 2005; Valin et al., 2016). This cleavage is associated with very low to
low grade metamorphism (Bastida et al., 1999; García-Lopez et al, 2018), and is overprinted by contact
metamorphic aureoles related to granodiorites emplacement at 292 + 2/− 3 Ma (Valverde-Vaquero et al., 1999).
**6. Discussion**



### 6.1 The geometry of the IAA

Most models for the IAA agree that the arc has a very tight geometry (Fig. 1). This interpretation assumes: 1) the correlation of stratigraphy of the northern limb of the arc across the Bay of Biscay, and 2) eastward continuation of the northern arm of the arc across NE Iberia. However, both assumptions are problematic.

1) The correlation across the northern arm across the Bay of Biscay is speculative due to poor exposure, and thus strongly depends on the location attributed to the Iberian Plate in Late Permian–Early Jurassic times. However the Mesozoic evolution of the Iberian Plate is controversial. Different interpretations of the Mesozoic Atlantic and Bay of Biscay magnetic anomalies, especially magnetic anomalies older than M0 chron (ca. 125 Ma), have yielded several end-member models for the reconstruction of the Iberian Plate motion with different initial positions (see Barnett-Moore et al. 2016 and Muñoz 2019 for a detailed discussion) (Fig. 9). This is a critical uncertainty as the position and orientation of the Iberian Plate at the end of the Variscan orogeny strongly constrains the geometry of the Ibero-Armorican Arc. Until the extent of rotation due to Alpine orogenesis is understood, the problem of lateral correlation involving the megastructures on both sides of the Bay of Biscay renders any reconstruction of the Ibero-Armorican Arc highly speculative.

2) The geometry of the southwestern arm of the Ibero-Armorican Arc is tightly constrained by correlations of the Cambrian-Ordovician sequences within WALZ and Ollo de Sapo magmatic rocks within the CIZ (Montero et al., 2007; Montero el al., 2009, among others), both of which can be traced for hundreds of kilometres along strike (Martínez-Catalán et al., 2007). However, the geometry of the eastern arm of the arc is not so well constrained. This eastern branch would include the Basque massif or sub-domain (western Pyrenees) and the south-westernmost inlier of the Coastal Catalonian Ranges (Priorat massif), which are thought to represent the lateral continuation of the West Asturian-Leonese Zone (Martínez-Catalán et al., 2007) (Fig. 1). In the same way, the remainder of the Pyrenees is thought to be equivalent to the CIZ (Martínez-Catalán et al., 2007) (Fig. 1). However, as widely documented, the Cambrian–Ordovician geodynamic, stratigraphic and zircon provenance evolution of the eastern Pyrenees fits better with other neighbouring areas, such as the Montagne Noire and southern Sardinia, than with the rest of the Iberian Massif (see discussion in Álvaro et al., 2018; Casas et al., 2019). As a result, the northern and eastern prolongations of the Ibero-Armorican Arc are not well defined, compromising interpretations of the proposed tight geometry of the arc.

### 6.2 The origin of the arc

The analysis of the $D_3$ structures affecting the autochthon of the GTMZ provided shortening values of about 5-10% (Fig. 6). This value is far less than the estimated 91%-94% of contraction required to form IAA as a secondary orocline, as proposed in the oroclinal and strike-slip models (Weil et al., 2000, 2001, 2010, 2013a, 2013b, 2019; Gutiérrez-Alonso et al., 2012; Fernández-Lozano et al., 2016; Pastor-Galán et al., 2011, 2012b, 2015a, 2017, 2019; Shaw et al., 2012; Martínez-Catalán et al. 2007; Martínez-Catalán, 2011). It is also significantly less than the 54% contraction implied in the model involving indentation and subsequent buckling (Casas and Murphy 2018).

In the CZ core of the arc, the horizontal contraction of 150±15 km related to the emplacement of Picos de Europa Unit (Merino-Tomé et al., 2009) must be re-evaluated. Using the control points proposed by these authors (Fig. 11, Merino-Tomé et al., 2009), the contraction related to the emplacement of the imbricate thrust system is around 58 km. These 58 km represent only one third of the approximate contraction of 164 km experienced by this segment of the Cantabrian Zone if it developed as a result of the bending around a vertical axis of a linear orogen (see insert in Fig. 4A). In addition, the age of emplacement of these youngest thrust units in the Cantabrian zone towards the



south is documented to have been ongoing from 304 to 299 Ma (Valin et al., 2016; Merino-Tomé et al., 2009),
coetaneous with the sedimentation of the molasses, Kasimovian to Gzhelian in age (Merino-Tomé et al., 2009,
2019-Fig. 11.5). In turn, these molasses unconformably overlie NW-SE and W-E-oriented thrusts and folds in the
southern Cantabrian Zone (e.g. in the Villablino coalfield, which cross-cuts the WALZ-CZ boundary (IGME, 1982),
and the La Magdalena coalfield (IGME, 1984). That is, these strata postdate not only the emplacement of the major
tectonic units in the CZ but also the arc formation, which must be older than the emplacement of the Picos de
Europa Unit. Moreover, the CIZ-WALZ boundary should had experienced a contraction of 109 km (32%), in a N-S
direction, during the closure of the arc (see insert in Fig. 4A), whereas no structures attributable to this have been
reported inside the WALZ.
Additionally, we propose that N-S contraction during the development of the Alpine deformation along the northern
Iberian margin may also have contributed to some tightening of the arc. In the southern border of the CZ, Marín et
al. (1995) estimated 20 km of contraction in Domo de Valsurvio and Curavacas syncline during the Alpine
deformation. Pulgar et al. (1999) have proposed a similar South displacement of ca. 22 km for the entire CZ. This
moderate Alpine overprinting of Variscan features is similar to that described in other areas of the Iberian Massif.
For example, Alpine thrusts caused displacements ranging between 25 to 30 km in the Iberian Range (Casas-Sáinz,
1993; Guimerà et al., 1995), and ca. 25 km in the Central System (Warburton and Álvarez, 1989). An Alpine
contraction of 40 km to 60 km in a NNE-SSW direction has been estimated for the whole Iberian Range (Guimerà et
al., 1995, 2004; Guimerà, 2018). The superposition of Alpine over Variscan folds has also been proposed in the
Carboniferous rocks of the Priorat area in the Catalan Coastal Range (Valenzuela et al., 2016).
Even considering late Variscan deformation in combination with Alpine deformation there is not enough contraction
to explain the formation of the arc as a secondary structure. The results of our analysis suggest the Ibero-Armorican
Arc is mainly a primary or non-rotational curvature, slightly modified during late Carboniferous and Alpine times.
This model implies that the relationship between the Central Iberian Arc and the Ibero-Armorican Arc, considered
both as margin-controlled curves, remains an open question.

**6.3 The role of the large strike-slip faults**

In the models involving strike-slip faulting deformation and oblique collision (Martínez-Catalán et al., 2007;
Martínez-Catalán, 2011) or indentation and subsequent buckling (Casas and Murphy, 2018), the large late-Variscan
dextral strike-slip faults play an important role (Fig. 2C and D). In both cases it has been assumed that the Porto
Tomar shear zone merges in the Armorican Massif with the South Armorican shear zones (Fig. 1) and that these
shear zones were initially linear. It is also assumed that they constitute the southern margin of the French Massif
Central. Using Crozon (Brittany) and Buçaco (Portugal) areas as piercing points, Casas and Murphy (2018)
estimated a total dextral offset of around 900 km between the Iberian and Armorican massifs during the late
Moscovian-Kasimovian along the Porto Tomar-South Armorican shear zone system, before its folding during the
arc deformation. However, if a primary origin for the arc is assumed, another relationship between these strike-slip
faults must be envisaged, as the South Armorican and the Porto Tomar shear zone system would not represent the
same megastructure before the arc formation. An alternative explanation is to consider that they constitute two
separate faults that acted consecutively. First, a dextral displacement along the NE-SW Porto Tomar shear zone of
about 450 km, and then offset of this structure by the dextral movement of the South Armorican shear zone system
oriented WNW-ESE. In this case, the northern prolongation of the Porto Tomar shear zone may be located
somewhere in the Northern Armorican Domain. The restoration of this strike-slip faults consecutive movement





results in a pre-tectonic arrangement of the different tectono-stratigraphic units compatible with the paleogeographic
reconstruction proposed by Casas and Murphy (2018).
**7. Conclusions**
From the analysis proposed here, it follows that late Variscan deformation together with the Alpine deformation is
not sufficient to explain the formation of Ibero-Armorican Arc as a secondary structure by means of vertical axis
rotations. We propose that this arc is mainly a primary, or non-rotational curve, slightly modified by ca. 10% of
superposed contraction during late Carboniferous and/or Alpine times.
Therefore, this amount of deduced contraction cannot account for the tight curvature of the Ibero-Armorican Arc if
originated as a linear orogenic feature. Moreover, the North side of this curvature is not supported by regional
geological data. We consider it necessary to re-evaluate the geometry of the arc, after considering how Alpine
movements affected the post-Variscan position of the Iberian Plate.
Another relationship between the late-Variscan large strike-slip faults in the Iberian and in the Armorican massifs
must be envisaged. Probably the Porto Tomar and the South Armorican shear zone systems do not represent the
same megastructure on both sides of the Bay of Biscay. Instead, these structures may have acted consecutively.
**Acknowledgements**
We acknowledge the financial support provided by CGL2017-87631-P, CGL2016-76438-P, PGC2018-093903-B-
C22 and SALTCONBELT-CGL2017- 85532-P projects, funded by Agencia Estatal de Investigación (AEI) and
Fondo Europeo de Desarrollo Regional (FEDER); and by project 2014SGR-467 (GEOMODELS Research Institute
and the Grup de Geodinàmica i Anàlisi de Conques). IDS thanks the financial support given by the "Estímulo ao
Emprego Científico – Norma Transitória" national science contract in the Faculdade de Ciências da Universidade de
Lisboa. This work is a contribution to the IGCP project 648, the IDL's Research Group 3 (Solid Earth dynamics,
hazards and resources) and to IDL's FCT-projects FCT/UID/GEO/50019/2019-IDL and FCT/UIDB/50019/2020-
IDL. Detailed revision by J.B. Murphy greatly improved a first version of this manuscript.

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





**FIGURE CAPTIONS**

**Fig. 1.** Map of the Variscan belt of Western and Central Europe at post-Variscan times (ca. 295 Ma, Early Permian) with the location of the different areas and structures referred to in the text, after Martínez-Catalán (2011) modified. Abbreviations: B (Buçaco); BCSZ (Badajoz-Córdoba shear zone); C (Crozon); CO (Corsica); CIZ (Central Iberian Zone); CZ (Cantabrian Zone); ECM (External crystalline massif of the Alps); FMC (French Massif Central); GTMZ (Galicia Tras-Os-Montes Zone); JPSZ (Juzbado-Peñalba shear zone); LC (Lizard Complex); LLF (Layale-Lubine fault); LT (Leon terrane shear zone); MAD (Mid Armorican Domain); MDZ (Moldanubian Zone); MM (Maures Massif); MS (Moravo-Silesian Unit); NAD (North Armorican Domain); NASZ (North Armorican shear zone); NEF (Nort-sur Erdre fault); OMZ (Ossa Morena Zone); PY (Pyrenees); PTSZ (Porto-Tomar shear zone); RHZ (Rheno Hercynian Zone); SA (Sardinia); SAD (Southern Armorican Domain); SAF (Southern Armorican front); SASZ (South Armorican shear zone, N and S: northern and southern branches); SISZ (Southern Iberia shear zone); SPZ (South Portuguese Zone); SXZ (Saxo-Thuringian Zone); TBZ (Teplá-Barrandian Zone); VF (Variscan Front); VM (Vosgues Massif); WALZ (West Asturian Leonese Zone). See text for explanation.

**Fig. 2.** Various models for the formation of the Ibero-Armorican Arc. A) Indentor model after Sánchez-García et al. (2003). B) Oroclinal bending about a vertical axis after Pastor-Galán et al. (2017). C) Dextral mega-shear model from Martínez-Catalán et al. (2007). D) Combination of margin-controlled and buckling from Casas and Murphy (2018).

**Fig. 3.** Contraction related to 90º rotations of both arms of an initially linear orogen with different initial lengths.

**Fig. 4**. Contraction related to: A) oroclinal bending about a vertical axis of an initial linear orogeny (after Pastor-Galán et al., 2017 modified); B) litospheric bending as a response to strike-slip faulting (after Martínez-Catalán et al., 2007 modified); C) arc formed as a result of combination of margin-controlled curve and buckling (after Casas and Murphy, 2018 modified).

**Fig. 5**. Estimation of the amount of lost lithosphere related to: A) oroclinal bending about a vertical axis of an initial linear orogeny (after Pastor-Galán et al., 2017 modified); B) litospheric bending as a response to strike-slip faulting (after Martínez-Catalán et al., 2007 modified); C) arc formed as a result of combination of margin-controlled curve and buckling (after Casas and Murphy, 2018 modified).

**Fig. 6.** A) Simplified geological map of the eastern rim of the Morais Allochthonous Complex (Galicia-Trás-os-Montes Zone) and the underlying parautochthonous and autochthonous domains, showing the main Variscan structures and the location of the cross section in B. This area is located in the axial zone of the Central Iberian Arc (CIA). B) Cross section cutting the axial zone of the CIA showing the horizontal shortening related to the $D_3$ structures in the autochthon of the CIZ, underlying the GTMZ. The contraction produced by $D_3$ folds at the base of the Armorican Quartzite (red thick lines) is measured in two sectors. Modified after Dias da Silva (2014) and Dias da Silva et al. (2020). Abbreviations: BLPD - Basal Lower Parautochthon Detachment; MTMT - Main Trás-os-Montes Thrust. For location, see Fig. 1.

**Fig. 7.** A) Geological map of the Ponga Unit that is at the core of the Cantabrian Zone arc, from Alvarez-Marrón (1995) (location in the inset). B) Axial traces of folds grouped into three sets frontal, corner, and lateral folds by Alvarez-Marrón (1995). The fault-bend folds have varied dimensions and orientations depending on the dimensions and orientations of the ramps and flats of the thrusts. Corner folds form oblique to the other two sets, at the intersection of lateral and frontal ramps. The interference and tectonic superposition of lateral and frontal thrust





structures cause the plunge of fold axis (see Alvarez-Marron, 1995 for the classification of the different types of fold
interactions). Frontal folds plunges are in the direction perpendicular to the transport direction. Lateral folds over
frontal ramps plunges are in the direction of the tectonic transport. For location, see Fig. 1.
**Fig. 8.** A) Distribution of footwall ramps and flats in the Ponga Unit basal thrust is shown as an example of how
large folds such as Rio Color Antiform and Rio Monasterio Antiforms are related to the lateral footwall ramps of the
basal thrust. Arrows indicate the emplacement direction of different thrust units mostly based on field kinematic
indicators (Alvarez-Marrón, 1995).B) Geological cross-section that illustrated the geometries of large E-W folds, the
thrust stacking and the lateral ramps in the footwall to the basal thrust (location in 7A). C) Block diagramme
showing the ideal geometry of the basal thrust of the Ponga Unit (Alvarez-Marrón, 1995). For location, see Fig. 1.
**Fig. 9.** End-member plate models reconstructions of the Iberian Plate relative to a fixed Eurasia at chron M0. A)
Taken from Srivastava et al. (2000); B) From Jammes et al. (2009), after Barnett-Moore et al. (2016) modified. Red
lines: Variscan dextral strike-slip faults, blue lines: Variscan sinistral strike-slip faults. Latitudes, in degrees, refer to
their present position in France.





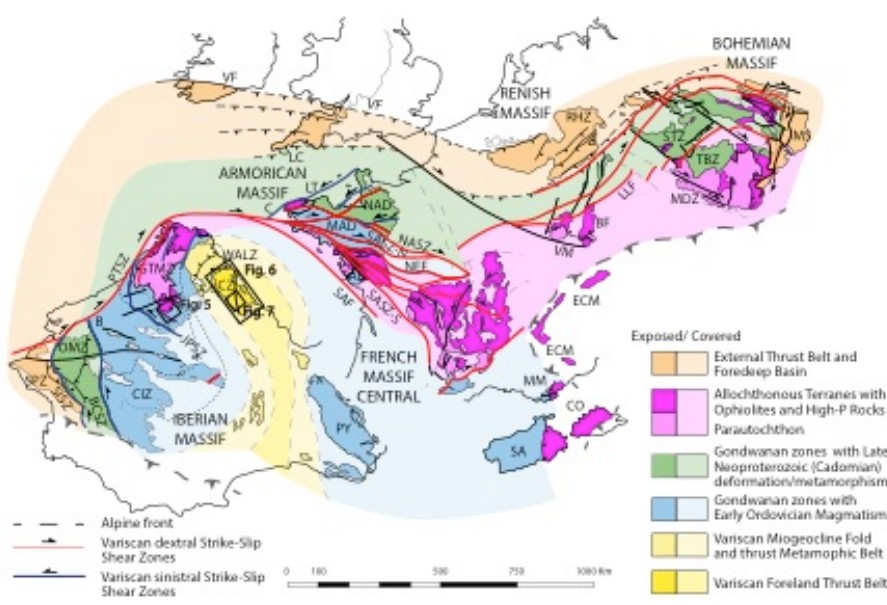



**Fig. 1.**



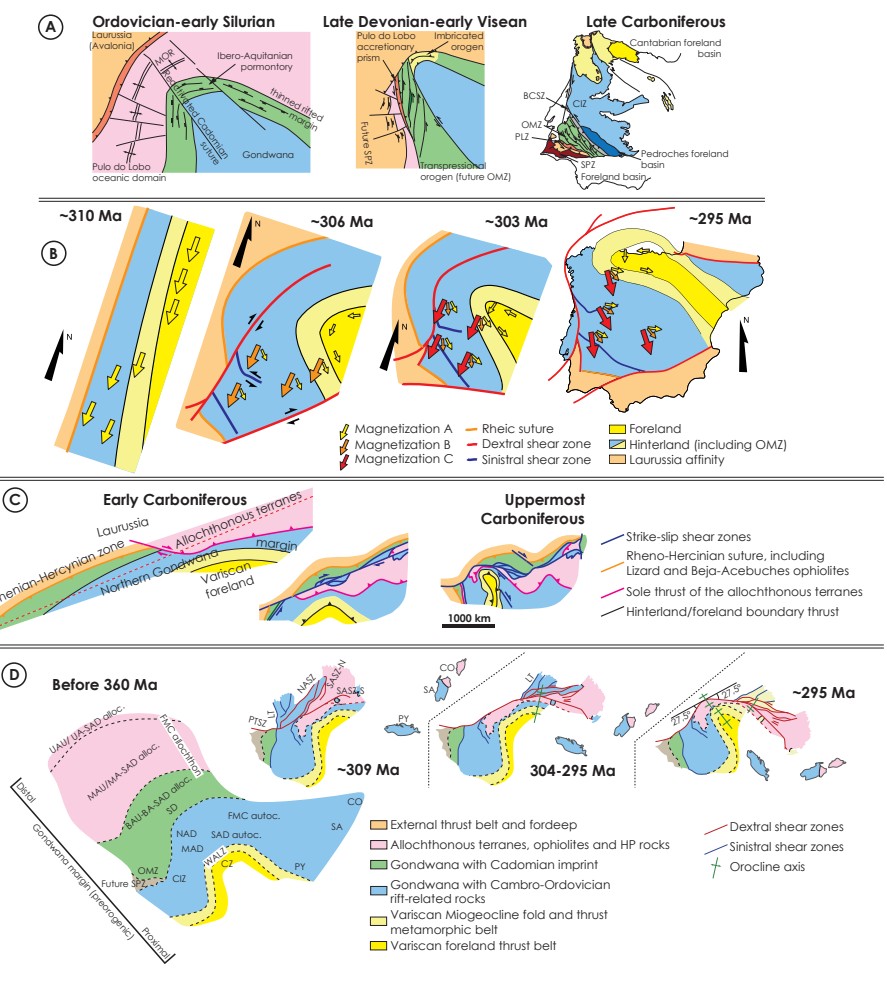



**Fig. 2.**





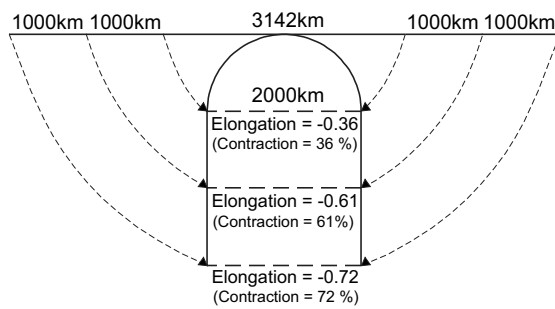



**Fig. 3.**



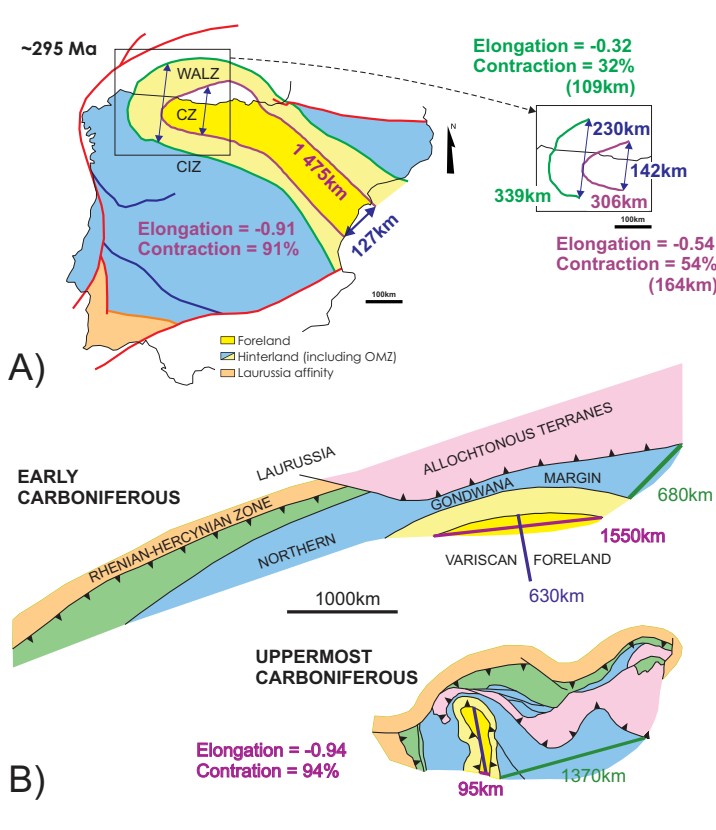

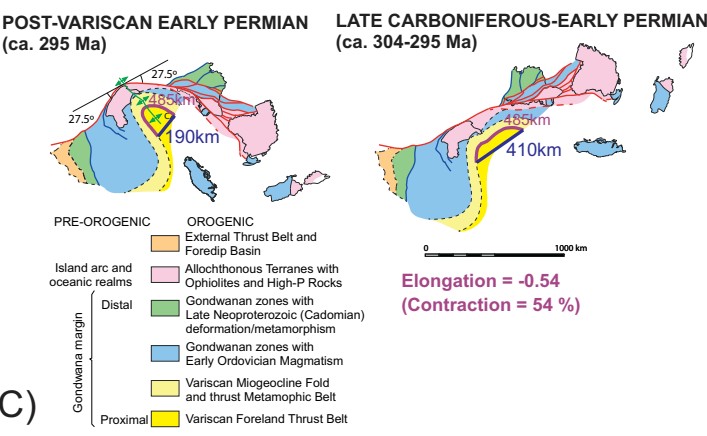



**Fig. 4.**





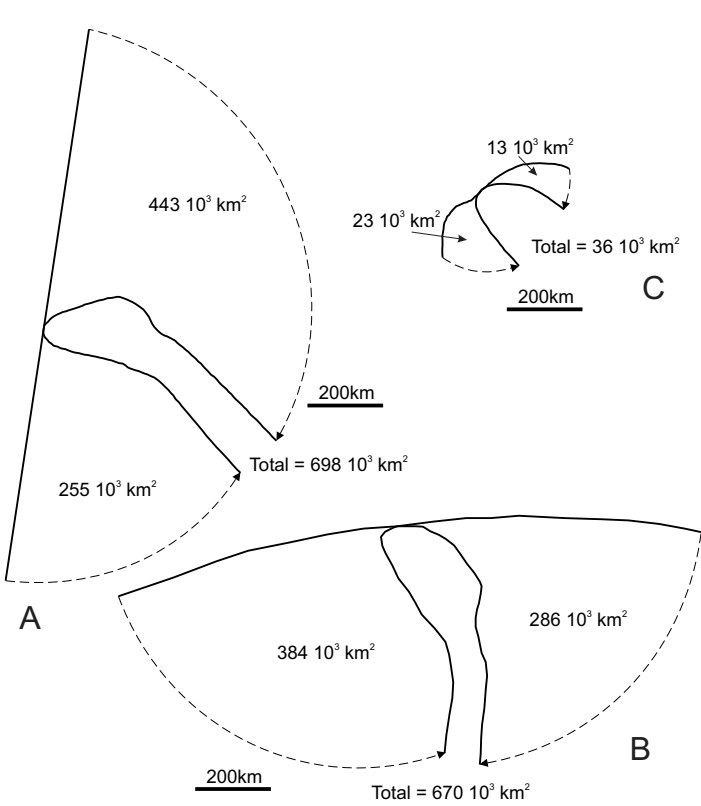



**Fig. 5.**





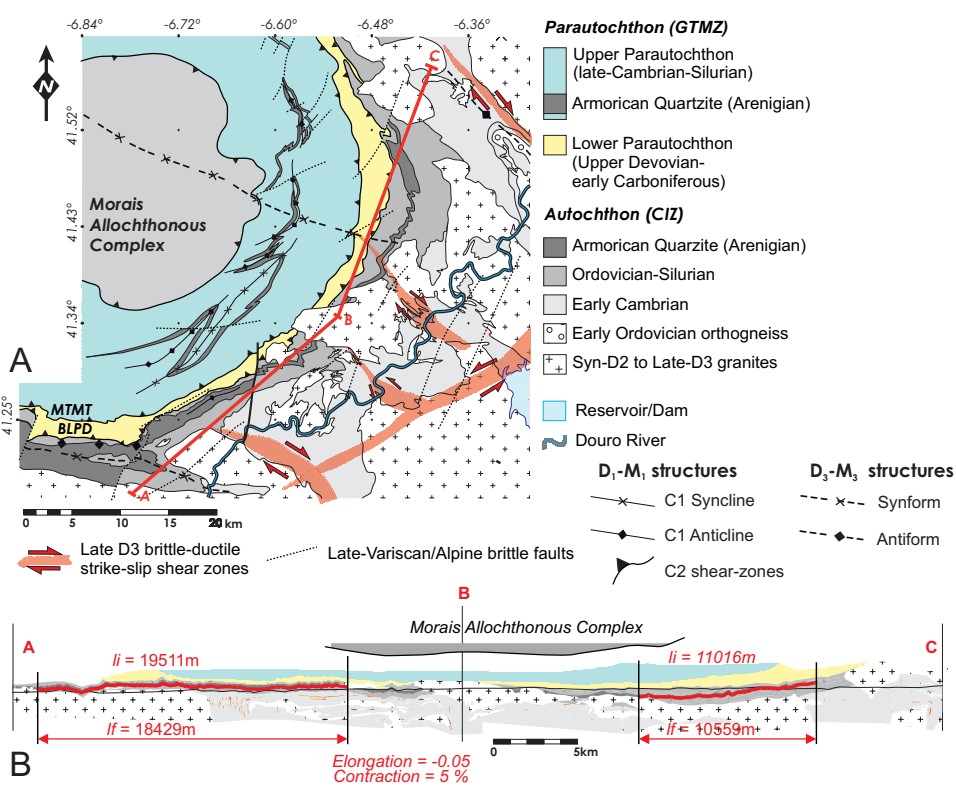



**Fig. 6.**





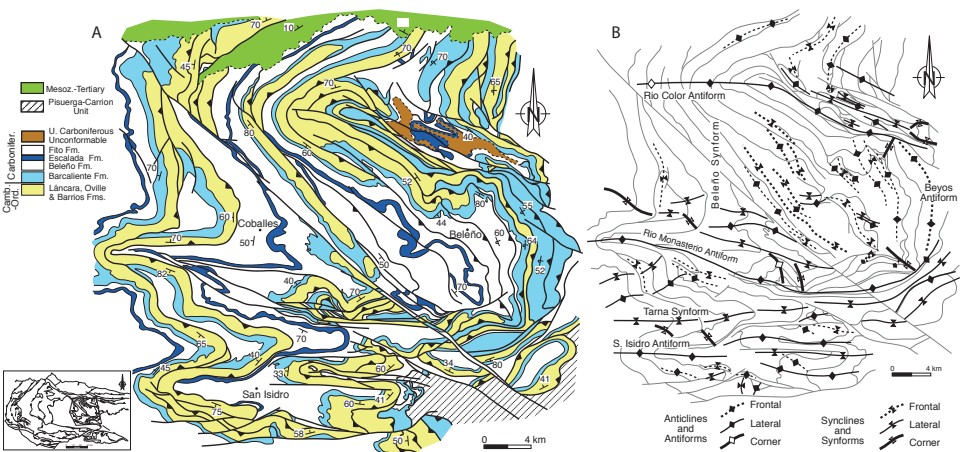



**Fig. 7.**





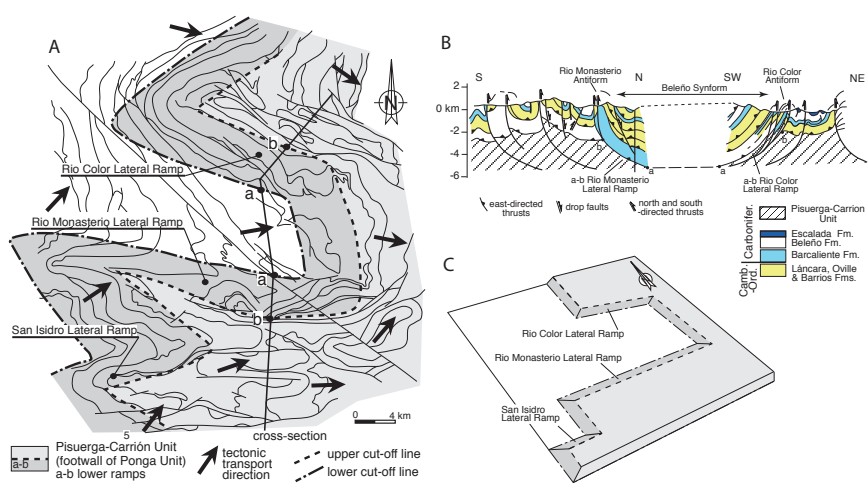



**Fig. 8.**





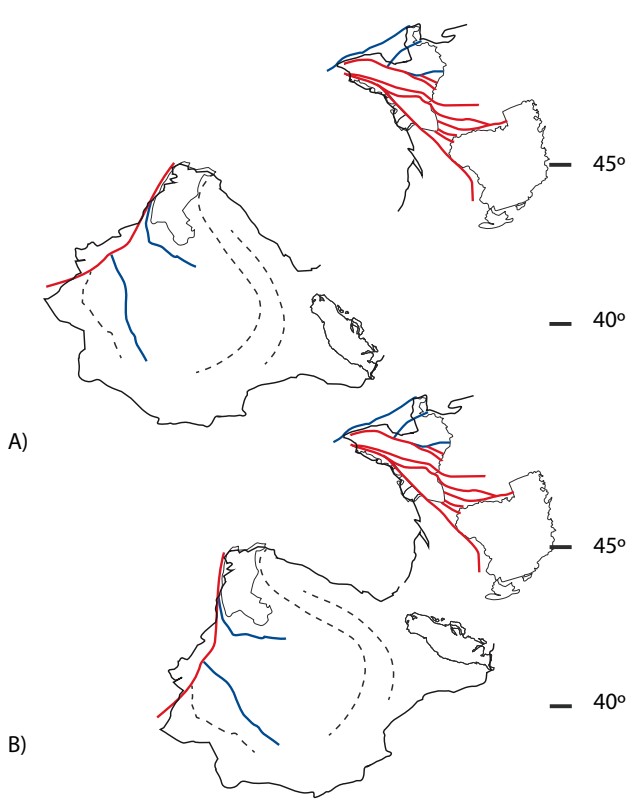



**Fig. 9.**
