# Peer review of "A first estimation of the contraction related to vertical axis rotation: the case of the Ibero-Armorican Arc formation"

_Solid Earth, 2020_

## Short Comment (SC1) · 20 Sep 2020

The Cantabrian arc and its oroclinal origin has been discussed since many times, using different techniques and approaches. The approach presented in this manuscript is very interesting and is essentially based on the calculation of the amount of shortening necessary for an orocline process. On this base the Author suggest that the amount of rotation is essentially less than previously proposed, and the present-day curvature of the arc is mostly related to the geometry of the Variscan foreland, and, therefore, that the arc is a primary one. This approach is innovative and interesting, but it seems to me that it is conflicting with paleomagnetic evidences that prove large amount of

paleomagnetic rotations, much more than the 25° proposed in the model. In fact, the oroclinal model proposed by many authors for this curved orogen is mainly based on paleomagnetic results, which fully supported this model.

I think that this is an essential point that should be carefully discussed before publishing the paper. In particular I think that a reliable model should explain both the amount of shortening and the amount of paleomagnetic rotation

---

## Short Comment (SC2) · 21 Sep 2020

Dear Massimo, Thank you very much for your comments. Yes, paleomagnetic data have been used as an evidence for the orocline model. However, it should be noted that they are not easy to interpret. Some of the involved rocks in the southern arm of the arc have not provided interpretable results, and in the other hand the results obtained differ in both branches of the arc. As you noted, the aim of our contribution in not discussing this data. We are dealing with geometrical aspects referred to contraction related to the arc formation. However, some points concerning paleomagnetic data can be considered:

[Figure]

1) The paleomagnetic results are quite different in the northern branch of the arc, in the core of the arc (Cantabrian zone) and in the southern (western) arm of the arc. In the northern branch, a ca. 25° clockwise rotation is proposed by Pastor-Galán et al. (2015b) to form the arc. In the central area, a post-Variscan folding and pre-orocline formation remagnetization suggests that the arc formation is due to late Kasimovian-Moscovian-Gzhelian rotation linked to an important reactivation of previously formed N-S oriented structures and the formation of radial folds and E-W oriented thrusts in the core of the arc (Weil et al. 200, 2001, 2010, 2012 and 2013). However, the results of the southern arm are more difficult to interpret. According to Pastor-Galán et al (2015a, 2016 and 2017) and Fernández Lozano et al. (2016) paleomagnetic declination vectors exhibit a wide dispersion, ranging from 60° (Fernández Lozano et al. 2016) to 90° (Pastor-Galán et al. 2017). Moreover, results obtained differ, depending on the type of analysed lithologies. These authors attribute the results to a remagnetization synchronous with the formation of the arc (Late Kasimovian-Early Permian). This interpretation has some important consequences: a) it implies that remagnetization processes were active for a long time interval (ca. 13 my, 310-297 Ma) in the southern branch of the arc, b) it implies a different timing for the remagnetization in the northern arm and in the core (previous to the arc formation), compared to the southern arm (synchronous with the arc formation), and c) it imposes a different kinematics for the formation of the arc, as in its northern arm the arc form as a result of 25° clockwise rotation, whereas in the southern arm a counter-clockwise rotation ranging from 70-90° is required (Pastor-Galán et al. 2015a, 2016 and 2017; Fernández Lozano et al. 2016). A closer view of the paleomagnetic results suggests that although this dispersion exists, when the data are considered grouped in their sites, the dispersion may be minimized. For instance, in Fig. 6 of Pastor-Galán et al (2017) the dispersion of the mean values of the sites is around 40°. Moreover, the deviation of the mean value of all the obtained vectors (138°/12.5°) from the Early Permian reference declination orientation (158°) is only ca. 20° (Figs. 8 and 12, Pastor-Galán et al. 2017).

2) In our opinion the most important point is that the paleomagnetic results of this south-

ern arm are not in accordance with regional geology data. Proposed counter-clockwise vertical axis rotations ranging from 70-90° implies shortening of several hundreds of kilometres and internal deformation in the southern arm of the arc to acquire an isoclinal fold geometry from an initial arc formed by tangential longitudinal strain (Weil at al. 2013). However, as we discuss in our contribution, no deformational structures accounting for these deformations are described, the remagnetization is post-Variscan folding (Pastor-Galán et al. 2015a, 2016), and these areas are characterized by simple structures with open upright folds and gently plunging fold axes (Pastor-Galán et al. 2016). Such simple structural arrangement allows Pastor-Galán et al. (2017) to discard structural complexities as the main source of scatter of the declination vectors in the southern arm of the arc. It should be noted, however, that the discrepancy between regional and paleomagnetic data is not discussed in paleomagnetic papers dealing with the southern arm of the arc. As our starting point is different, from regional geology data, we cannot use paleomagnetic data that are not in accordance with the regional data, to constrain our proposed reconstruction.

We are conscious that a detailed discussion of the points outlined above is beyond the scope of this paper. In future contributions, we intend to discuss in detail various aspects of this complex geology.

Best regards,

Josep Maria Casas

---

## Referee Comment (RC1) · Anonymous Referee #1 · 23 Sep 2020

This manuscript is related with the Ibero-Armorican Arc and its title suggests that estimations of the contraction related to vertical axis rotation would be presented. This sound very interesting, nevertheless, it fails as –in my opinion– these estimations are very, very rough and are not presented rigorously. Moreover, the manuscript is not well structured, discussion is mixed with geological setting and the data are not clearly presented. Specific comments are summarized below: "1. Introduction (Lines 27 to 60)": Unless the lector know very well the cited papers, the introduction is difficult to follow. The authors should summarize in a few words what is necessary to know for the manuscript understanding. I am surprised that the term of "progressive arc" does not appear at any moment of the introduction, as it is one of the main models of arc

formation, together with orocline and primary arc models. "2. Geological setting of the Iberian Massif (lines 61 to 138)": Again, it is very difficult to follow the manuscript; many of the terms are not localized and there are a lot of details which are not useful for the remaining of the manuscript. I would suggest to the author to make a table with summarized of the characteristics, both lithostratigraphic and structural, of each one of the arc (or arcs, if they include the second arc some authors propose, the Central Iberian Arc of Martínez-Catalan et al. 2014). That really would help the lector to understand the controversy and to distinguish what is the contribution of the authors to solve it. "3. Proposed models for the arc formation (Lines 139 to 194)": In this epigraph, previous models are discussed. The problem is that they are discussed before the new data are presented in the manuscript. I would propose to the authors to distinguish between: a) the observations with which everybody agree and to put them in the geological setting, and b) those which are not accepted by the scientific community, which should be discussed in a discussion epigraph, after the presentation of the author data. "4. The amount of contraction required by the different proposed models (Lines 195 to 222)": The "simplified way to estimate the amount of contraction" (line 196) is so oversimplified that it is not serious enough (see also figs. 3 and 5). This is a rough estimation that one can write on a piece of paper, but not in a scientific manuscript. My suggestion is that this rough estimation, which can be a nice introduction to more precise data should be accompanied by more precise estimations which include not only the type of structures and the shortening due to each set of structures, but also the timing of these latter. This is a key point which should be addressed. "5. The geological data (lines 223 to 318)": In this paragraph, we come back to the geological setting (from lines 224 to 262 and 269 to 318) accompanied with a few data (lines 263 to 268) based on an unlocalized map (or does it corresponds with the square with Figure 5 label in Figure 1?). The authors must absolutely follow the classical structure of any scientific manuscript, differentiating clearly geological setting, methodology, results, discussion. "6. Discussion (lines 319 to 401)": At the beginning of the discussion (lines 321 to 323), as far as I understand, the authors wonder about the existence of the Ibero-Armorican

Arc. In that case, I don't understand why they are making estimation of the contraction related to vertical axis rotation. They should have begun with the discussion of these correlation problems. This is very confused. The remaining discussion is also difficult to follow, as they cite of a lot of localities and case studies not localized in the manuscript figures. "7. Conclusions": In my opinion, according to what I exposed in the previous paragraphs, the conclusions are not supported by the data.

For all these reasons, the manuscript should be rejected in its present form and be totally rewritten and reorganized following the comments I made, if possible.

──────────────────────────────

---

## Editor Comment (EC1) · Federico Rossetti (Editor) · 27 Sep 2020

The SC1 is an invited review. Accordingly, the interactive discussion step for the review of the manuscript se-2020-126 can be closed with the RC1 and the SC1.

Federico Rossetti

---

## Author Comment (AC1) · 8 Oct 2020

Dear reviewer, Thank you very much for your comments. We agree that the manuscript may be improved following some of your suggestions. To the best of our knowledge, "progressive arc" models have not been invoked for the formation of the Ibero-Armorican arc. Maybe the most similar could be the indentor model, which is thought to originate a progressive deformation in the indented plate. We will consider this point in the revised version of the manuscript. In the same way, we will reorganise the Introduction in order to clarify which is the main problem we would like to address. In our view, this is that the different models proposed for the arc formation do not consider the geo-

metrical consequences of their proposals. This is particularly evident when considering the amount of contraction required for the different vertical axis rotations proposed for the formation of the Ibero-Armorican Arc. Concerning your comment about the method we use: "these estimations are very, very rough and are not presented rigorously", we have to say that we present a first estimation of this contraction. As far as we know, nobody has tried this approach before. The proposed geometry of the Variscan Arc is at the scale of hundreds of kilometers, defined by the boundaries between the Variscan zones in Iberia. The only markers that can be used to estimate its deformation at that hecto-kilometric scale are those boundaries. Deformation at smaller scales should be consistent to this analysis, if a secondary arc is to be accepted. Our analysis is not, then, an oversimplification. Moreover, we think that our conclusion is that the deformation observed from structures at smaller scales than that of the arc is far less than the one needed to explain its formation from a previous linear orogen, and even we found some inconsistences on the proposed age of the arc development. So, the statement that "not only the type of structures and the shortening due to each set of structures, but also the timing of these latter" should be asked to the authors who proposed that the Variscan Arc is secondary. Concerning the surface measurement methods, we would like to precise that in order to estimate the amount of surface of lost lithosphere needed –assuming the Ibero-Armorican Arc was formed as an orocline or a secondary fold forming as a result of strike-slip faulting– the original maps were escalated in a CAD environment (Microstation$^®$). The boundaries of the lost area were defined comparing the WALZ-CZ boundary previous and after the arc formation, and assuming an arcuate path to the line tips during deformation. The areas bounded between these three lines were measured using the CAD tool for this purpose and rounded to 103 km2. In this estimation, the values of lost lithospheric surface should be considered as minimum, as it is assumed that there is no change in the position of the fold hinge during its development In a general way, we agree that the Geological Setting is hard to read for anyone not familiar with the complex geology of the Iberian Massif. We will try to make it more clear and readable in the new version and also to improve the location

of the localities and case studies not localized in the figures. In the same sense, we will rewrite and reorganize the manuscript in order to expose our ideas in a more clear way. Thank you again for your comments. Sincerely yours, On behalf of the co-authors

Josep Maria Casas

---

## Editor Comment (EC2) · Federico Rossetti (Editor) · 9 Oct 2020

Dear Authors,

Your manuscript is a potentially interesting contribution. However, both reviewers' reports indicate that the manuscript is not suitable for publication as it stands. Much work is needed to improve its internal structure and to better focus the scientific rationale. It is necessary to integrate the proposed reconstruction in the vast background information dealing with orocline formation and, in particular, with the available paleomagnetic data for the region. Proposal of an alternative model should be consistent with the paleomagnetic evidence of block rotations or, alternatively, implemented with an exhaustive

discussion and critical reassessment of the paleomagnetic data set. Submission of a revised version is thus only encouraged if the above points will be carefully considered during manuscript revision. In any case, the revised manuscript will be subject to a further revision round.

Federico Rossetti

---

## Author Comment (AC2) · 13 Oct 2020

Dear editor, Thank you very much for your comments, but unfortunately we are not able to access to the second referee's report. We can only read the post of M. Mattei (20 Sep 2020), and the one of the Anonymous Referee #1 (23 Sep 2020). So, we don't know the details of the second referee comments. Please, let us know how we can get to this report. 1) Concerning your comments, we like to explain further our reasons for not doing a discussion and critical reassessment of the paleomagnetic data set in our work: 1.1) This data set supports only one of the proposed models published that explain the origin of the Ibero-Armorican arc, that is the arc formed as a result

of the rotation about a vertical axis of an initial linear orogen. In our contribution we also present other different competing models, but we don't go into the geological arguments supporting the different proposals because our starting point is different. We focus on the deformation needed for those vertical axis rotations from the estimation of the amount of contraction (horizontal shortening) and the amount of surface of lost lithosphere needed assuming the Ibero-Armorican Arc was formed as an orocline or a secondary fold. 1.2) The coherence between the available paleomagnetic data for the region and any proposed model is a main point. But, as explained above, paleomagnetic data should be considered, as well as any other data that can be significant in an orogen scale. On the contrary, most paleomagnetic analyses have not considered the regional-deformational implications of huge rotations around a vertical axis at the scale analysed. So, in our opinion if we follow your suggestion the relationship between paleomagnetic and regional geological studies usually is not well balanced. The latter ones have to discuss and consider the palomagnetic data, but the former ones usually do not take into account the regional implication of their proposals. 1.3) As stated, in our paper we deal with this point and conclude that the deformation needed for those vertical axis rotations is not found. Then, a contradiction appears that should not be solved by considering only one type of data. This should be one of the main conclusions of our work. 1.4) Some points concerning the internal coherence of the paleomagnetic data that have been already published previously (Casas and Murphy, 2018). We included a brief summary in the response to the M. Mattei's comments: It should be noted that the paleomagnetic data are not easy to interpret. Some of the involved rocks in the southern arm of the arc have not provided interpretable results, and in the other hand the results obtained differ in both branches of the arc. The paleomagnetic results are quite different in the northern branch of the arc, in the core of the arc (Cantabrian zone) and in the southern (western) arm of the arc. In the northern branch, a ca. 25° clockwise rotation is proposed by Pastor-Galán et al.(2015b) to form the arc. In the central area, a post-Variscan folding and pre-orocline formation re-magnetization suggests that the arc formation is due to late Kasimovian-Moscovian-

[Figure]

Gzhelian rotation linked to an important reactivation of previously formed N-S oriented structures and the formation of radial folds and E-W oriented thrusts in the core of the arc (Weil et al. 200, 2001, 2010, 2012 and 2013). However, the results of the southern arm are more difficult to interpret. According to Pastor-Galán et al (2015a, 2016 and 2017) and Fernández Lozano et al. (2016) paleomagnetic declination vectors exhibit a wide dispersion, ranging from 60°(Fernández Lozano et al.2016) to 90° (Pastor-Galán et al. 2017). Moreover, results obtained differ, depending on the type of analysed lithologies. These authors attribute the results to a re-magnetization synchronous with the formation of the arc (Late Kasimovian-Early Permian). This interpretation has some important consequences: a) it implies that re-magnetization processes were active for a long time interval (ca. 13 my, 310-297 Ma) in the southern branch of the arc, b) it implies a different timing for the re-magnetization in the northern arm and in the core (previous to the arc formation), compared to the southern arm (synchronous with the arc formation), and c) it imposes a different kinematics for the formation of the arc, as in its northern arm the arc form as a result of 25° clockwise rotation, whereas in the southern arm a counter-clockwise rotation ranging from 70-90° is required (Pastor-Galán et al. 2015a, 2016 and 2017; Fernández Lozano etal. 2016). A closer view of the paleomagnetic results suggests that although this dispersion exists, when the data are considered grouped in their sites, the dispersion may be minimized. For instance, in Fig. 6 of Pastor-Galán et al (2017) the dispersion of the mean values of the sites is around 40°. Moreover, the deviation of the mean value of all the obtained vectors (138°/12.5°) from the Early Permian reference declination orientation (158°) is only ca. 20°(Figs. 8 and 12, Pastor-Galán et al. 2017). In our opinion the most important point is that the paleomagnetic results of this southern arm are not in accordance with regional geology data. Proposed counter-clockwise vertical axis rotations ranging from 70-90° implies shortening of several hundreds of kilometres and internal deformation in the southern arm of the arc to acquire an iso-clinal fold geometry from an initial arc formed by tangential longitudinal strain (Weil et al. 2013). However, as we discuss in our contribution, no structures related to these deformations are described,

the re-magnetization is post-Variscan folding (Pastor-Galán et al. 2015a, 2016), and these areas are characterized by simple structures with open upright folds and gently plunging fold axes (Pastor-Galán et al.2016). Such simple structural arrangement allows Pastor-Galán et al. (2017) to discard structural complexities as the main source of scatter of the declination vectors in the southern arm of the arc. As stated, it should be noted, however, that the discrepancy between regional structure and paleomagnetic data is not discussed in paleomagnetic papers dealing with the southern arm of the arc. As our starting point is different, from regional geology data, we cannot use paleomagnetic data that are not in accordance with the regional data to constrain our proposed reconstruction. We are conscious that a detailed discussion of the points outlined above is beyond the scope of this paper. We think that a detailed discussion of the various aspects of this complex geology is a matter for a paper by itself.. 2) We agree with the Anonymous Referee #1 that the manuscript may be improved following some of his suggestions. 2.1) To the best of our knowledge, "progressive arc" models have not been invoked for the formation of the Ibero-Armorican arc at the scale of the orogen. Maybe the most similar could be the indentor model, which is thought to originate a progressive deformation in the indented plate. We will consider this point in the revised version of the manuscript. In the same way, we will reorganise the Introduction in order to clarify which is the main problem we would like to address. 2.2) Concerning his comment about the method we use: "these estimations are very, very rough and are not presented rigorously", we have to say that we present a first estimation of this contraction. As far as we know, nobody has tried this approach before. The proposed geometry of the Variscan Arc is at the scale of hundreds of kilometers, defined by the boundaries between the Variscan zones in Iberia. The only markers that can be used to estimate its deformation at that hecto-kilometric scale are those boundaries. Deformation at smaller scales should be consistent to this analysis, if a secondary arc is to be accepted. Our analysis is not, then, an oversimplification. 2.3) Moreover, we think that our conclusion is that the deformation observed from structures at smaller scales than that of the arc is far less than the one needed to explain its formation from a previous

linear orogen, and even we found some inconsistences on the proposed age of the arc development. So, the statement that "this rough estimation . . . should be accompanied by more precise estimations which include not only the type of structures and the shortening due to each set of structures, but also the timing of these latter" should be asked to the authors who proposed that the Variscan Arc is secondary. 2.4) Concerning the surface measurement methods, we would like to precise that in order to estimate the amount of surface of lost lithosphere needed –assuming the Ibero-Armorican Arc was formed as an orocline or a secondary fold forming as a result of strike-slip faulting–, the original maps were escalated in a CAD environment (Microstation®). The boundaries of the lost area were defined comparing the WALZ-CZ boundary previous and after the arc formation, and assuming an arcuate path to the line tips during deformation. The areas bounded between these three lines were measured using the CAD tool for this purpose and rounded to 103 km2. In this estimation, the values of lost lithospheric surface should be considered as minimum, as it is assumed that there is no change in the position of the fold hinge during its development. 2.5) In a general way, we agree that the Geological Setting is hard to read for anyone not familiar with the complex geology of the Iberian Massif. We will try to make it more clear and readable in the new version and also to improve the location of the localities and case studies not localized in the figures. In the same sense, we will rewrite and reorganize the manuscript in order to expose our ideas in a more clear way. Thank you again for your comments and we await your decision considering our comments in order to submit the revised version in the due time. Sincerely yours, On behalf of the co-authors,

Josep Maria Casas

References: Casas, J.M.; Murphy B. (2018) Unfolding the arc: The use of pre-orogenic constraints to assess the evolution of the Variscan belt in Western Europe. Tectonophysics, 736: 47-61. Fernández-Lozano, J., Pastor-Galán, D., Gutiérrez-Alonso, G., and Franco, P.: New kinematic constraints on the Cantabrian orocline: A paleomagnetic study from the Peñalba and Truchas synclines, NW Spain, Tectonophysics, 681, 195-

208, doi: https://doi.org/10.1016/j.tecto.2016.02.019, 2016. Pastor-Galán, D., Groenewegen, T., Brouwer, D., Krijgsman, W., and Dekkers, M. J.: One or two oroclines in the Variscan orogen of Iberia? Implications for Pangea amalgamation, Geology, 43, 527-530, doi: 10.1130/g36701.1, 2015a Pastor-Galán, D., Ursem, B., Meere, P. A., and Langereis, C.: Extending the Cantabrian Orocline to two continents (from Gondwana to Laurussia). Paleomagnetism from South Ireland, Earth and Planetary Science Letters, 432, 223-231, doi: 10.1016/j.epsl.2015.10.019, 2015b Pastor-Galán, D., Dekkers, M. J., Gutiérrez-Alonso, G., Brouwer, D., Groenewegen, T., Krijgsman, W., Fernández-Lozano, J., Yenes, M., and Álvarez-Lobato, F.: Paleomagnetism of the Central Iberian curve's putative hinge: Too many oroclines in the Iberian Variscides, Gondwana Research, 39, 96-113, doi: 10.1016/j.gr.2016.06.016, 2016 Pastor-Galán, D., Gutiérrez-Alonso, G., Dekkers, M. J., and Langereis, C. G.: Paleomagnetism in Extremadura (Central Iberian zone, Spain) Paleozoic rocks: extensive remagnetizations and further constraints on the extent of the Cantabrian orocline, Journal of Iberian Geology, doi: 10.1007/s41513-017-0039-x, 2017 Weil, A. B., Van der Voo, R., van der Pluijm, B. A., and Parés, J. M.: The formation of an orocline by multiphase deformation: a paleomagnetic investigation of the Cantabria–Asturias Arc (northern Spain), Journal of Structural Geology, 22, 735-756, doi: 10.1016/S0191-8141(99)00188-1, 2000 Weil, A. B., Van der Voo, R., and Van der Plujim, B. A.: Oroclinal bending and evidence against the Pangea megashear: The Cantabria-Asturias arc (northern Spain), Geology, 29, 991-994, doi: 10.1130/0091-7613(2001)029<0991:OBAEAT>2.0.CO;2, 2001 Weil, A. B.: Kinematics of orocline tightening in the core of an arc: Paleomagnetic analysis of the Ponga Unit, Cantabrian Arc, northern Spain, Tectonics, 25, doi: 10.1029/2005tc001861, 2006 Weil, A. B., Gutiérrez-Alonso, G., and Conan, J.: New time constraints on lithospheric-scale oroclinal bending of the Ibero-Armorican Arc: a palaeomagnetic study of earliest Permian rocks from Iberia, Journal of the Geological Society, 167, 127-145, doi: 10.1144/0016-76492009-002, 2010 Weil, A. B., Gutiérrez-Alonso, G., Johnston, S. T., and Pastor-Galán, D.: Kinematic constraints on buckling a lithospheric-scale orocline along the northern margin of Gondwana: A geologic synthesis, Tectonophysics, 582, 25-49, doi: 10.1016/j.tecto.2012.10.006, 2013a Weil, A. B., Gutiérrez-Alonso, G., and Wicks, D.: Investigating the kinematics of local thrust sheet rotation in the limb of an orocline: a paleomagnetic and structural analysis of the Esla tectonic unit, Cantabrian–Asturian Arc, NW Iberia, International Journal of Earth Sciences, 102, 43-60, doi: 10.1007/s00531-012-0790-3, 2013b Weil, A., Pastor-Galán, D., Johnston, S. T., and Gutiérrez-Alonso, G.: Late/Post Variscan Orocline Formation and Widespread Magmatism. In: The Geology of Iberia: A Geodynamic Approach: Volume 2: The Variscan Cycle, Quesada, C. and Oliveira, J. T. (Eds.), Springer International Publishing, Cham, doi: 10.1007/978-3-030-10519-8_14, 2019